# A Taxonomic Study on the Genus *Naddia* from China (Coleoptera, Staphylinidae, Staphylininae) with Descriptions of Two New Species [note 1]

**DOI:** 10.3390/insects13060503

**Published:** 2022-05-26

**Authors:** Mei-Hua Xia, Liang Tang, Harald Schillhammer

**Affiliations:** 1Department of Biology, Shanghai Normal University, 100 Guilin Road, 1st Educational Building 423–A Room, Shanghai 200234, China; 18255536512@163.com; 2Naturhistorisches Museum Wien, Burgring 7, A-1010 Wien, Austria; harald.schillhammer@nhm-wien.ac.at

**Keywords:** Staphylinini, *Naddia*, identification key, rove beetle

## Abstract

**Simple Summary:**

*Naddia* is a genus of rove beetles distributed in Asia particularly in its south-eastern part. The species of the genus usually have a large body size and can be recognized by the emarginated base of the head. *Naddia* species are found to be mimics of sphecid or vespid wasps in both behavior and appearance, which is reported in this paper for the first time. With two new species described in the present paper, 39 species are known from the world, and 12 species are known from China.

**Abstract:**

Two new species of *Naddia* from China are described: *N. chenchangchini* sp. nov. (Guangxi) and *N. hujiayaoi* sp. nov. (Zhejiang, Fujian, Jiangxi, Hunan, Guangxi, Guangdong). *Naddia atripes* Bernhauer, 1939 is new to Vietnam and the following Chinese provinces: Jiangxi, Guangxi, Jiangsu, Shaanxi, Hunan, Hubei, Sichuan, Guizhou, Fujian and Guangdong. *Naddia chinensis* Bernhauer, 1929 is new to Zhejiang, Jiangsu, Guizhou, Henan, Hunan, Shaanxi, Fujian, Chongqing, Guangxi and Hubei. *Naddia miniata* Fauvel, 1895 is new to Laos. A key to Chinese species of the genus is provided.

## 1. Introduction

*Naddia* Fauvel, 1867 is a strictly Asian genus of the rove beetle subtribe Staphylinina. The genus can be readily distinguished from other members of Staphylinina by the deeply (rarely moderately) emarginated base of the head and a characteristic aedeagus with a flat and spoon-shaped apical portion of the median lobe in most species [1,2], some species having a weakly sclerotized subapical portion. The paramere is variable in size, from gladiate to very inconspicuous in shape, with a tiny rudiment at the aedeagal base. A few setae are always situated at the tip of the paramere, and sometimes they are arranged in two groups. The length of the setae can be used as character to distinguish similar species, e.g., in *N. chinensis* and *N. miniata*. A complex of sclerites exists in the aedeagal inner sac, although it is indistinct in some species.

Presently, 37 species of the genus have been described worldwide [3,4,5], 11 of them having been recorded from China: *N. atripes* Bernhauer, 1939 from Zhejiang [6]; *N. chinensis* Bernhauer, 1929 from Beijing, Gansu [7,8]; *N. hainanensis* Yang & Zhou, 2010 from Hainan [8]; *N. ishiharai* Shibata, 1994 from Taiwan [9]; *N. malaisei* Scheerpeltz, 1965 from Yunnan [10]; *N. mangshanensis* Yang & Zhou, 2010 from Fujian, Hunan, Guangdong [8]; *N. miniata* Fauvel, 1895 from Jiangxi, Sichuan, Yunnan [8,11]; *N. monticola* Shibata, 1994 from Taiwan [9]; *N. nanlingensis* Yang & Zhou, 2010 from Guangdong [8]; *N. rufipennis* Bernhauer, 1915 from Yunnan [12]; *N. taiwanensis* Shibata, 1979 from Taiwan [13,14]. Among them, the record of *N malaisei* from China is problematic. The record of *N. malaisei* from Yunnan can be only found in the Catalogue of Palaearctic Coleoptera (Volume 2) [15] without any traceable references. The species was originally described from southern Shan-States of Myanmar based on a female. The entry in the catalog was most likely based on a misconception, as most species in the Scheerpeltz paper dealing with Malaise’s Myanmar material have been described from Kambaiti on the Yunnan border. A few species, including possible undescribed species, are very similar to *N. malaise**i* in appearance. So, we prefer to remove the Chinese record of *N. malaise**i* for now. *Naddia chinensis* and *N. miniata* are extremely similar to each other in appearance and their distributions published in the previous paper [8] are partially incorrect, which has been corrected in the present paper. Additionally, two new species, *N. chenchangchini* sp. nov. and *N. hujiayaoi* sp. nov., are described. Thus, the total number of *Naddia* species is increased to 39, and the number of Chinese species is increased to 12.

## 2. Materials and Methods

The specimens examined in this paper were collected individually from the ground, by sifting leaf litter and by flight intercept traps and pitfall traps. They were subsequently killed with ethyl acetate. For examination of the genitalia, the last three abdominal segments were detached from the body after softening in hot water. The aedeagus together with other dissected pieces were mounted in Euparal (Chroma Gesellschaft Schmidt, Koengen, Germany) on plastic slides beneath the carded specimens. Photos of sexual characteristics were taken with a Canon G9 camera attached to an Olympus SZX 16 stereoscope; habitus photos were taken with a Canon macro photo lens MP-E 65 mm attached to a Canon EOS 7D camera and stacked with Zerene Stacker (http://www.zerenesystems.com/cms/stacker, accessed on 12 May 2022).

The specimens treated in this study are deposited in the following collections:CFEN—Private collection of Zhen-Hao Feng, Nanjing, ChinaCSHUB—Collection of Michael Schülke, in “Museum der Alexander Humboldt Universität”, Berlin, GermanyNMW—Natural History Museum Wien, AustriaSHNU—Department of Biology, Shanghai Normal University, Shanghai, China

Body measurements are abbreviated as follows:
BL—body length, measured from the anterior margin of the clypeus to the posterior margin of abdominal tergite XFL—forebody length, measured from the anterior margin of the clypeus to the apex of the elytra (apicolateral angle)HL—length of head along the midlineHW—width of head including eyesCL—length of eyePO—length of post-ocular regionPL—length of pronotum along the midlinePW—width of pronotum at the widest pointEL—length of elytra, measured from humeral angleEW—width of elytra at the widest point

## 3. Results


**
*Naddia miniata*
**
**Fauvel, 1895**


*Naddia miniata* Fauvel, **1895**, 250; Yang & Zhou, **2010**, 9.

**Material examined. China: Yunnan:** 1♂, 1♀, Dehong Pref., Mengyang Town, Mangxuan Mount., 31 October 2021. (SHNU); 1♂, 1♀, Mengla, Menglun, alt. 600 m, 17 March 2018, Y.-Q. Lu leg. (SHNU); **Laos: Viangchan:** 2♂♂, Phou Khao Khouay Nat. Park, Nam Leuk, env. Tad Leuk Waterfall, 1–8 June 1996, alt. 200 m, FIT, H. Schillhammer leg. (NMW); **Hua Phan:** 1♂, Ban Saluei, Phou Pan, 20°12′ N 104°01′ E, alt. 1500–1900 m, 23 April–15 May 2008, C. Holzschuh leg. (NMW); 1♂, same, but 10 May–15 June 2009 (NMW); 4♂♂, 3♀♀, same, but 1–31 May 2011 (NMW); 3♂♂, 6♀♀, same, but 11 April–15 May 2012 (NMW); 1♂, same, but 3–30 April 2014 (NMW); **Xieng Khouang:** 1♀, 30 km NE Phonsavan, Nong Pet—Phou San, 19°34.657′ N, 103°23.313′ E, alt. 1300 m, 3 August 2008, S. Tarasov leg. (NMW).

**Measurements**. Male: BL: 18.6–18.9 mm, FL: 10.6–10.8 mm. HL: 3.34–3.61 mm, HW: 3.45–3.67 mm, CL: 1.17 mm, PO: 1.50–1.61 mm, PL: 3.34–3.45 mm, PW: 3.34 mm, EL: 4.17–4.28 mm, EW: 4.00–4.28 mm. HW/HL: 1.00–1.02, PO/CL: 1.29–1.38, PL/PW: 1.00–1.03, EL/EW: 1.00–1.04.

Female: BL: 19.7 mm, FL: 11.1 mm. HL: 3.61 mm, HW: 3.73 mm, CL: 1.17 mm, PO: 1.61 mm, PL: 4.00 mm, PW: 3.61 mm, EL: 4.45 mm, EW: 4.17 mm. HW/HL: 1.03, PO/CL: 1.38, PL/PW: 1.11, EL/EW: 1.07.

**Distribution**. China (Yunnan), Myanmar, Laos. New to Laos.

**Diagnosis.** The species was originally described from southern Myanmar. In general appearance (Figure 1A,B), it is extremely similar to *N. chinensis*, which was later described from China. The differences between *N. miniata* and *N. chinensis* were previously mentioned in the shape of head or body [7,8]. Dr. Aleš Smetana, who examined the types of both species, could not detect reliable differences between them and supposed that *N. chinensis* should be synonymized with *N. miniata* (personal communication). By examining large series of specimens with similar appearance from China in various collections, the specimens from Yunnan and Laos are found to be different from other specimens by the much longer setae on the paramere (Figure 2A–D). Considering the geographical distribution, the specimens from Yunnan are regarded as *N. miniata* in this paper. In addition, the elytral pubescence of *N. miniata* is more reddish. The distributional records of Jiangxi and Sichuan for *N. miniata* in the previous paper [8] are assigned to the distribution of *N. chinensis*.


***Naddia chinensis* Bernhauer, 1929**


*Naddia chinensis* Bernhauer, **1929**, 110; Yang & Zhou, **2010**, 11.

**Material examined. China: Jiangxi:** 1♂, Mt. Sanqing, alt. 700–1000 m, 04 May 2005, Hu & Tang leg. (SHNU); 1♂, Yichun City, Fengxin County, Baizhang Vill, alt. 1000–1300 m, 16 July 2013, Hu & Lv leg. (SHNU); **Shaanxi:** 1♂, Lueyang, Wulongdong, alt. 1038 m, 12 May 2021, Juan, Li. et al. leg. (SHNU); 1♂, Baoji City, Jiulongdong, alt. 1083 m, 26 May 2021, Juan Li et al. leg. (SHNU); 1♀, Hanzhong, Tiantaishan, alt. 995 m, 13 July 2021, Juan, Li. et al. leg. (SHNU); 1♀, Shou Man vill., 32°14′ N, 108°34′ E, alt. 1000–1700 m, 15 June–15 July 2000 (no collector mentioned) (CSHUB); **Chongqing:** 1♀, Daba Shan, 15km S Shou Man vill., 32°08′ N, 108°37′ E, alt. 1800 m, 25 May–14 June 2000, Siniaev & Plutenko leg. (CSHUB); **Jiangsu:** 1♂, Nanjing City, Mt. Zijin, alt. 311 m, 20 January 2020, Ming-Yuan Fan leg. (SHNU); 1♀, Nanjing City, July 2012, Cong-Wu Feng leg. (SHNU); **Sichuan:** 1♂, Yingjing County, Daxiangling, alt. 1200 m, 02 July 2009, Hao Huang leg. (SHNU); 1♀, 70 km W Chengdu, Qinchenhou mts., alt. 1400 m, 21–25 June 2005, S. Murzin leg. (CSHUB); 2♂♂, 1♀, 5 km NW Wenchuan, alt. 1500 m, 27 May–9 June 2006, A. Puchner leg. (NMW); **Henan:** 1♂, Xinyang, Huangbaishan Mt., Lemei Farmhouse, alt. 730 m, 03 September 2021, H. Liu leg. (SHNU); **Hunan:** 1♂, Yanling Co., Nanfengmian, mixed forest, leaf litter, wood sifted, alt. 1380 m, 27 May 2014, Peng, Shen, Yu & Yan leg. (SHNU); **Fujian:** 1♀, Fuzhou City, 27 April 2005, Yue Liu leg. (SHNU); 1♂, Guadun [“Kuatun”], 22 July 1946, Tschung Sen leg. (NMW); 1♀, Guadun [“Kuatun”], 16 May 1946, Tschung Sen leg. (NMW); 2♀♀, Guadun [“Kuatun”], 16 May 1946, Tschung Sen leg. (NMW); 1♂, 2 km SE Xinqiao, 27.05°N, 117.1°E, 10–29 May 2005, J. Turna leg. (NMW); 1♂, Shiniushan, 25°38′ N, 118°28′ E, alt. 1600–1700 m, 1–28 May 2008, J.Turna leg. (NMW); **Hubei:** 1♀, Xuanen City, Changtanhe County, Qizimeishan, alt. 1172 m, 1–2 January 2015, Qiu & Xu leg. (SHNU); **Guangxi**: 1♂, 1♀, Nanning City, Mt. Dawangling, Jiangxi zhen, 16–29 May 2014, at light, Wang leg. (NMW); **Guizhou:** 1♂, Guiyang City, Nanming sheltern park, alt. 1000 m, February 2020, Shi-Yu Zha leg. (SHNU); 1♂, Mt. Fenghuangshan, alt. 900 m, 06 March 2012, Run-Yu Li leg. (SHNU); 1♀, Guiyang City, Tuyunguan, alt. 1189 m, 21 June 2015, Qiu & Xu leg. (SHNU); 2♀♀, Leishan Co., Leigong Shan, east slope, 26°26.59′ N, 108°16.53′ E, Fangxian vill. env., 13–24 June 2001, H. Schillhammer leg. (NMW); **Zhejiang:** 1♂, Lin’an City, Mt. Tianmu, 15–28 August 2003, Hu & Tang leg. (SHNU); 1♂, Lin’an City, Mt. Tianmu, alt. 250 m, 18 August 2010, Jia-Yao Hu leg. (SHNU); 1♂, Qingliangfeng N. R., Longtangshan, alt. 1100 m, 12 May 2012, Chen, Ma & Zhao leg. (SHNU); 1♂, Lin’an City, Mt. West Tianmu, 11–22 August 2011, Hong-Ye Lai leg. (SHNU); 1♂, Lin’an City, Tianmushan W., alt. 300 m, 30 May 2014, Tang leg. (SHNU); 1♀, Lin’an City, West Tianmushan, alt. 300 m, 17 August 2013, Su-Nan Huang leg. (SHNU); 1♀, Jinhua City, Pan’an County, Dapanshan, July 2015, Su-Jiong Zhang leg. (SHNU); 2♀♀, Lin’an City, West Tianmu shan, 21 August 2008, Jia-Yao Hu leg. (SHNU); 1♀, Lin’an City, Mt. West Tianmu, 11–22 August 2011, Hu & Tang leg. (SHNU); 1♀, Mt. W. Tianmu, 15–28 August 2005, Mei-Jun Zhao leg. (SHNU); 1♀, Ningbo City, Tiantong, 15 July–15 August 2021, HE Zhu-Qing leg. (SHNU).

**Measurements**. Male: BL: 16.4–20.0 mm, FL: 10.0–11.1 mm. HL: 3.11–3.61 mm, HW: 3.34–3.61 mm, CL: 1.00–1.22 mm, PO: 1.45–1.67 mm, PL: 3.34–4.17 mm, PW: 3.34–3.73 mm, EL: 4.34–4.45 mm, EW: 4.17–4.45 mm. HW/HL: 0.97–1.07, PO/CL: 1.24–1.50, PL/PW: 0.95–1.17, EL/EW: 1.00–1.07.

Female: BL: 16.7–22.2 mm, FL: 9.5–11.7 mm. HL: 3.50–4.00 mm, HW: 3.39–4.17 mm, CL: 1.11–1.33 mm, PO: 1.56–1.78 mm, PL: 3.34–4.00 mm, PW: 3.34–4.17 mm, EL: 3.89–5.00 mm, EW: 3.89–5.00 mm. HW/HL: 0.97–1.06, PO/CL: 1.33–1.50, PL/PW: 0.96–1.08, EL/EW: 0.93–1.14.

**Distribution**. China (Beijing, Gansu, Zhejiang, Jiangsu, Guizhou, Sichuan, Henan, Hunan, Shaanxi, Fujian, Jiangxi, Chongqing, Guangxi, Hubei). New to Zhejiang, Jiangsu, Guizhou, Henan, Hunan, Shaanxi, Chongqing, Guangxi, Fujian and Hubei.

**Diagnosis.** The species is extremely similar to *N. miniata*, both in appearance (Figure 1C,D) and in shape of the aedeagus. It may be distinguished from the latter by the shorter setae on the aedeagal paramere (Figure 2E–H) and the generally denser reddish pubescence on the elytra. The ratios of HW/HL are quite variable in *N. chinensis*; therefore, this characteristic is not very helpful in recognizing *N. miniata* and *N. chinensis*.


**
*Naddia taiwanensis*
**
**Shibata, 1979**


*Naddia taiwanensis* Shibata, **1979**, 19; Hu, **2020**, 344.

**Material examined. China: Taiwan:** 1♀, Hepinglindao, alt. 900 m, 16 May 2014, Y.-F. Chen leg. (SHNU)

**Measurements**. Female: BL: 19.2 mm, FL: 11.7 mm. HL: 3.34 mm, HW: 3.45 mm, CL: 1.22 mm, PO: 1.50 mm, PL: 3.89 mm, PW: 3.73 mm, EL: 4.45 mm, EW: 4.45 mm. HW/HL: 1.03, PO/CL: 1.23, PL/PW: 1.04, EL/EW: 1.00.

**Distribution**. China (Taiwan).

**Diagnosis.** The species was described from Taiwan without comparing it to *N. chinensis*, most likely because the author did not know the latter. In fact, *N. taiwanensis* is almost identical to *N. chinensis* except for the antennal segments 8–11, which are relatively shorter than those of *N. chinensis*. Examining more specimens of *N. taiwanensis* will be necessary to evaluate if it is a valid species.


**
*Naddia mangshanensis*
**
**Yang & Zhou, 2010**


*Naddia mangshanensis* Yang & Zhou, **2010**, 5.

**Material examined. China: Guangdong:** 1♂, Ruyuan County, Nanling N. R., Shikengkong, alt. 1820 m, 30 April 2015, Peng, Tu & Zhou leg. (SHNU); 2♂♂, Ruyuan County, Nanling N. R., Disilindao, alt. 1500 m, 05 May 2015, Peng, Tu & Zhou leg. (SHNU); 1♂, Shaoguan City, Guangdongdiyifeng, alt. 1538–1784 m, 28 June 2020, Xia, Zhang, Yin & Lin leg. (SHNU); **Hunan:** 1♀, Mangshan N. R., PT, 10 May 2020. (SHNU).

**Measurements**. Male: BL: 16.7–21.1 mm, FL: 9.5–10.6 mm. HL: 3.61–3.78 mm, HW: 3.84–3.89 mm, CL: 1.11 mm, PO: 1.56–1.95 mm, PL: 3.61–3.89 mm, PW: 3.22–3.50 mm, EL: 3.06–3.56 mm, EW: 3.34–3.89 mm. HW/HL: 0.92–1.06, PO/CL: 1.40–1.75, PL/PW: 1.03–1.17, EL/EW: 0.85–1.00.

Female: BL: 23.4 mm, FL: 12.2 mm. HL: 4.45 mm, HW: 4.11 mm, CL: 1.22 mm, PO: 2.17 mm, PL: 4.17 mm, PW: 3.73 mm, EL: 3.61 mm, EW: 4.17 mm. HW/HL: 0.93, PO/CL: 1.77, PL/PW: 1.12, EL/EW: 0.87.

**Distribution**. China (Fujian, Guangdong, Hunan).

**Diagnosis.** The brachypterous species is similar to three full-winged species, *N. miniata*, *N. chinensis* and *N. taiwanensis* in general appearance (Figure 3A,B), but it can be easily distinguished from those by shorter elytra and posterior margin of abdominal tergite VII without distinct palisade fringe.


***Naddia chenchangchini* Xia, Tang & Schillhammer sp. nov.**


**Type material. Holotype. China: Guangxi:** ♂, glued on a card with labels as follows: “China: Guangxi A. R., Nanning City, Mt. Daming, alt. 1200 m, 02 June 2014, Yan-Quan Wu leg.” “Holotype/*Naddia chenchangchini*/Xia, Tang & Schillhammer” [red handwritten label] (SHNU). **Paratypes. Guangxi:** 1♂, Nanning City, Mt. Damingshan, alt. 1200 m, 01 August 2012, Wei-Jun He leg. (SHNU); 1♀, Nanning City, Mt. Daming, 31 July 2012, Wen-Xuan Bi leg. (SHNU); 1♀, Nanning City, Mt. Daming, alt. 1200 m, 02 June 2014, Yan-Quan Wu leg. (NMW). All paratypes with yellow handwritten label: Paratype/*Naddia chenchangchini*/Xia, Tang & Schillhammer”.

**Description.** Measurements of male: BL: 21.2–23.6 mm, FL: 11.8–12.0 mm. HL: 4.17 mm, HW: 3.89–4.17 mm, CL: 1.11–1.22 mm, PO: 2.11–2.22 mm, PL: 4.17–4.28 mm, PW: 3.61 mm, EL: 3.89–4.17 mm, EW: 4.17–4.45 mm. HW/HL: 0.93–1.00, PO/CL: 1.73–2.00, PL/PW: 1.15–1.18, EL/EW: 0.93–0.94.

Measurements of female: BL: 23.4 mm, FL: 12.2–12.8 mm. HL: 4.45 mm, HW: 4.34–4.73 mm, CL: 1.22 mm, PO: 2.28–2.39 mm, PL: 4.17–4.45 mm, PW: 3.89–4.17 mm, EL: 3.89–4.45 mm, EW: 4.56–4.73 mm. HW/HL: 0.98–1.06 mm, PO/CL: 1.86–1.95, PL/PW: 1.07, EL/EW: 0.85–0.94.

Body blackish, elytra coppery with reddish pubescence, lateral portions of abdominal tergites III–VI with patches of golden pubescence (less distinct on tergite VI in one specimen), tergite VII with a patch of golden pubescence in basal half, mouthparts and tarsi brownish, antennae dark brown.

Head with lateral margins slightly divergent posteriad, dorsal surface with dense and contiguous punctation except on frons, where punctation is shallow and less dense; ventral surface densely punctate with large portions at posterior angles impunctate; antennae with club cylindrical, antennomeres 1–3 distinctly oblong, antennomere 2 shorter than antennomere 3, antennomeres 4–10 longer than wide, antennomere 11 longer than penultimate, antennomeres 8–10 slightly asymmetrical, antennomere 11 asymmetrical.

Pronotum widest at anterior angles, anterior angles distinctly prominent, lateral margins distinctly curved ventrad slightly in front of midlength; dorsal surface with punctures similar to those of head; an impunctate midline exists in posterior third.

Elytra with lateral margins slightly divergent posteriad, punctation very dense and fine. Scutellum densely and finely punctate, pubescence black.

Abdominal tergites with dense punctation and short pubescence.

Male. Sternite VIII with posterior margin emarginate in the middle; aedeagus (Figure 4E–H) symmetrical, median lobe elongated, middle portion weakly sclerotized, nearly transparent, inner sac with sclerotized sclerites; paramere very short, separated into two pieces with few apical setae.

Female. Sternite VIII with posterior margin entire.

**Distribution**. China (Guangxi).

**Diagnosis.** The new brachypterous species is extremely similar to *N. mangshanensis*, but it can be distinguished from the latter by in average larger size, abdominal tergite VI with longitudinal golden pubescent patches laterally, lobes of paramere more distinctly pointed and with longer apical setae (Figure 4E–H).

**Etymology.** This species is named in honor of Mr. Chang-Chin Chen, who donated *Naddia* specimens of the new species to us.


***Naddia nanlingensis* Yang & Zhou, 2010**


*Naddia nanlingensis* Yang & Zhou, **2010**, 3.

**Material examined. China: Guangdong:** 1♀, Ruyuan County, Nanling N. R., Laopengkeng, mixed forest, leaf litter, wood sifted, alt. 1360 m, 29 April 2015, Peng, Tu & Zhou leg. (SHNU); **Hunan:** 2♂♂, Mangshan N. R., PT, 10 May 2020. (SHNU).

**Measurements**. Male: BL: 17.8–18.3 mm, FL: 10.0–10.1 mm. HL: 3.34 mm, HW: 3.50–3.67 mm, CL: 0.89–0.95 mm, PO: 1.89–2.00 mm, PL: 3.61–3.67 mm, PW: 3.11–3.22 mm, EL: 3.06 mm, EW: 3.34–3.45 mm. HW/HL: 1.05–1.10, PO/CL: 2.00–2.25, PL/PW: 1.12–1.18, EL/EW: 0.89–0.92.

Female: BL: 17.8 mm, FL: 10.6 mm. HL: 3.50 mm, HW: 3.61 mm, CL: 0.83 mm, PO: 2.11 mm, PL: 3.61 mm, PW: 3.34 mm, EL: 3.34 mm, EW: 3. 45 mm. HW/HL: 1.03, PO/CL: 2.53, PL/PW: 1.08, EL/EW: 0.97.

**Distribution**. China (Guangdong, Hunan).

**Diagnosis.** The brachypterous species is somewhat similar to *N. mangshanensis* with overlapping distribution, but it can be easily distinguished from the latter by the following characteristics: elytra black without bronze tint, apical margin of terminal antennal segment markedly concave, forming an acute tip, ventral side of head without impunctate areas at posterior angles.


***Naddia atripes* Bernhauer, 1939**


*Naddia atripes* Bernhauer, **1939**, 597.

**Material examined. China: Zhejiang:** 1♂, Longquan, Fengyang Mt., Mihou Valley, alt. 950 m, 09 May 2019, Tang & Zhao leg. (SHNU); 1♂, Qingliangfeng N. R., Longtangshan, alt. 1100 m, 12 May 2012, Chen, Ma & Zhao leg. (SHNU); 1♂, Longquan City, Fengyangshan N. R., Datianping, alt. 1350 m, 30 April 2016, Jiang, Liu & Zhou leg. (SHNU); 1♂, Jiande City, Chunan County, Qiandaohu, alt. 110 m, July 2013, Song leg. (SHNU); 1♂, Pan’an County, Yuantang Linchang, alt. 1800 m, 16 July 2012, Su-Jiong Zhang leg. (SHNU); 1♀, Dapanshan, 11 July 2012, Su-Jiong Zhang leg. (SHNU);1♀, Pan’an, Dapanshan, Huangtan State-owned Forest Farm, alt. 1000 m, 04 April 2013, Su-Jiong Zhang leg. (SHNU);2♀♀, Jinhua City, Pan’an County, Dapanshan, July 2015, Su-Jiong Zhang leg. (SHNU);1♀, Anji City, Longwangshan Mt., alt. 300–500 m, 07 June 2012, Hu & Yin leg. (SHNU); 1♀, Ningbo City, Mt. Tiantong, 27 April 2019, Qin-Fen Sheng leg. (SHNU); 1♀, Jinhua City, Pan’an County, Dapanshan N. R., alt. 531–845 m, 06 May 2016, Jiang, Liu & Zhou leg. (SHNU); 1♂, Caoyutang For. Park, 27°55′ N, 119°39′ E, alt. 1100–1300 m, 5–6 May 2009, J. Turna leg. (NMW); 5♂♂, 1♀, same, but 3–31 May 2010, J. Turna leg. (NMW); **Shaanxi**: 1♂, 1♀, Shou Man vill., 32°14′ N, 108°34′ E, alt. 1000–1700 m, 15 June–15 July 2000 local collector (CSHUB); **Jiangxi:** 1♂, Nanchang City, Jiangxi Agricultural University, 06 May 2018, Yi-Yang Xu leg. (SHNU); 1♀, Mt. Wuyi, alt. 950 m, 10 May 2005, Hu & Tang leg. (SHNU); 4♂♂, Sanqingshan, 28°52′ N, 118°03′ E, alt. 600 m, 18 April–17 July 2007, J. Turna leg. (NMW); 5♂♂, Jiulian Shan Nat. Res., 24°32′ N, 114°28′ E, alt. 750 m, 27 May–20 June 2015, Jatua leg. (NMW); **Hubei**: 1♂, Mufu Shan, Jiugongshan For. Park., 29°24′ N, 114°36′ E, up to 1000 m, 3–18 June 2002, J. Turna leg. (NMW); 8♂♂, ~3 km S Duncun, 31°00′ N, 110°57′ E, 12 May–5 June 2005, J. Turna leg. (NMW) **Guangxi:** 1♂, Huaping, Cujiang Station, PIT, 18 May 2020. (SHNU); 4♂♂, Huaping, Cujiang Station, PIT, 09 May 2020. (SHNU); 8♂♂, Huanjiang, Jiuwan Mt., Yangmeiao, FIT, alt. 1250m, 25 April 2021, Tang, Peng, Cai & Song leg. (SHNU); **Jiangsu:** 1♂, Nanjing, Hongshan Zoo, 32°05′38″ N, 118°47′33″ E, 06 April 2022, alt. 17m, Zhen-Hao Feng leg. (CFEN); **Hunan:** 1♂, Mangshan N. R., FIT, 10. May 2020. (SHNU); 1♂, Mangshan N. R., PT, 10 May 2020. (SHNU); 1♂, Mangshan N. R., PT, alt. 846m, 20 May 2020. (SHNU); 2♀♀, Yanlin County, Taoyuandong N. R., Nanfengmian, alt. 1394m, 17 June 2016, Jiang, Liu, Jiang & Zhou leg. (SHNU); 3♂♂, Wuling Shan, Zhangjiajie, 29°24′ N, 110°24′ E, ca. 700 m, 4–7 July 2003, J. Turna leg. (NMW); **Sichuan**: 1♂, Emei Shan, alt. 1050–1150 m, 1–9 July 2008, A. Puchner leg. (NMW); **Guizhou**: 2♂♂ 1♀, Yaogu env., 25°20′ N, 107°56′ E, alt. 800–900 m, 1–13 June 2011, Jatua leg. (NMW); 10♂♂, Xiaoqikong, 25°15′ N, 107°41′ E, alt. 690 m, 9–29 June 2013, Jatua leg. (NMW); 7♂♂, 1♀, Xiaoqikong, 25°16′ N, 107°41′ E, alt. 870 m, 22 May–7 June 2016, Jatua leg. (NMW); 3♂♂, ESE Shuilixiang, 25°30′ N, 107°51′ E, alt. 960 m, 22 May–4 June 2018, Jatua leg. (NMW); **Fujian**: 2♀♀, Guadun [“Kuatun”], 12 July 1946 and 15 August 1946 [resp.], Tschung Sen leg. (NMW); 3♂♂, Fenshui Guan, 27°54′ N 117°51′ E, 1–4 June 2004, J. Turna leg. (NMW); 1♂, Taining, E env., 26°54′ N, 117°09′ E, 9–28 May 2005, J. Turna leg. (NMW); 1♂, ~2 km SE Xinqiao, 27°02′ N, 117°06′ E, alt. 640 m, 23 April–20 July 2006, J. Turna leg. (NMW); 2♂♂, Tianbaoyanshan, NW slope, 25°58′ N, 117°31′ E, 20 June–12 July 2007, J. Turna leg. (NMW); 10♂♂, Wuyi Shan, ~10 km W Xingcun, 27°40′ N, 117°48′ E, alt. 270 m, 8–25 May 2005, J. Turna leg. (NMW); 2♂♂, same, but 25 June–17 July 2007 (NMW); 2♂♂, Jiuxian Shan, 25°42′ N, 118°07′ E, alt. 1350 m, 25 June–17 July 2007, J. Turna leg. (NMW); 11♂♂, ~10 km E Yong’an, 25°58′ N, 117°27′ E, alt. 700 m, 28 April–31 May 2008, J. Turna leg. (NMW); 1♂, Mandangshan, 26°42′ N, 118°07′ E, alt. 460–900 m, 8 May–4 June 2010, J. Turna leg. (NMW); 9♂♂, Tiantaishan For. Park, 25°43′ N, 117°17′ E, alt. 1100–1200 m, 20 May–27 June 2011, J. Turna leg. (NMW); **Guangdong**: 1♂, Yunji Shan Nat. Res., 24°06–07′ N, 114°10′ E, alt. 700–1300 m, 13–23 June 2013, Jatua leg. (NMW); 2♂♂, Pingyun Shan, 22°00′ N, 111°10′ E, alt. 1000 m, 6–25 June 2015, Jatua leg. (NMW); 1♂, Dachou Ding, 24°16–17′ N, 112°24′ E, alt. 680–850 m, 10–30 June 2015, Jatua leg. (NMW); **Vietnam: Cao Bằng:** Pia Ouac Nat. Park, bel. Salmon Station, ca. 1360 m, 22°35’40” N, 105°53’22” E, 9–18 May 2019, FIT, Brunke & Schillhammer leg. [20A] (NMW).

**Measurements**. Male: BL: 15.0–20.0 mm, FL: 8.9–10.0 mm. HL: 2.84–3.06 mm, HW: 3.06–3.45 mm, CL: 0.95–1.06 mm, PO: 1.45–1.56 mm, PL: 2.78–3.34 mm, PW: 2.95–3.06 mm, EL: 3.61–3.89 mm, EW: 3.78–4.06 mm. HW/HL: 1.08–1.18, PO/CL: 1.37–1.65, PL/PW: 0.91–1.09, EL/EW: 0.96–1.00.

Female: BL: 13.3–18.3 mm, FL: 8.9–10.0 mm. HL: 2.78–3.17 mm, HW: 3.17–3.34 mm, CL: 0.89–1.06 mm, PO: 1.39–1.67 mm, PL: 2.78–3.34 mm, PW: 2.89–3.34 mm, EL: 3.61–4.17 mm, EW: 3.78–4.45 mm. HW/HL: 1.04–1.18, PO/CL: 1.37–1.67, PL/PW: 0.96–1.02, EL/EW: 0.93–1.00.

**Distribution**. The species is widely distributed in Southern and Southeastern China (Zhejiang, Jiangxi, Shaanxi, Guangxi, Jiangsu, Hunan, Hubei, Sichuan, Guizhou, Fujian, Guangdong). New to Vietnam and all Chinese provinces except Zhejiang.

**Diagnosis.** The species is very similar to *N. ishiharai* from Taiwan, but it can be distinguished from the latter by more asymmetrical antennal club and more distinct patch of silvery pubescence on abdominal tergites VI and VII (Figure 5A,B and Figure 6E–H).


**
*Naddia ishiharai*
**
**Shibata, 1994**


*Naddia ishiharai* Shibata, **1994**, 315.

**Material examined**. None.

**Distribution**. China (Taiwan).


***Naddia rufipennis* Bernhauer, 1915**


*Naddia rufipennis* Bernhauer, **1915**, 54.

**Material examined. China: Yunnan:** 1♂, Xishuangbanna, Jinghong City, Gasa Town, Nanpaxiaozhai, ca. 1100 m, 25 March 2021, Hui Ce leg. (SHNU).

**Measurements**. Male: BL: 14.3 mm, FL: 8.5 mm. HL: 2.50 mm, HW: 2.67 mm, CL: 0.95 mm, PO: 1.11 mm, PL: 2.78 mm, PW: 2.50 mm, EL: 3.22 mm, EW: 3.00 mm. HW/HL: 1.07, PO/CL: 1.18, PL/PW: 1.11, EL/EW: 1.07.

**Distribution**. China (Yunnan), Myanmar.

**Diagnosis.** This species can be easily recognized by its unique coloration.


***Naddia monticola* Shibata, 1994**


*Naddia monticola* Shibata, **1994**, 318.

**Material examined. China: Taiwan:** 1♂, Kaohsiung, Duonalindao, 10 September 2014, B.-X. Guo leg. (SHNU).

**Measurements**. Male: BL: 15.0 mm, FL: 8.3 mm. HL: 2.56 mm, HW: 2.67 mm, CL: 0.78 mm, PO: 1.39 mm, PL: 2.78 mm, PW: 2.61 mm, EL: 3.17 mm, EW: 3.34 mm. HW/HL: 1.04, PO/CL: 1.79, PL/PW: 1.06, EL/EW: 0.95.

**Distribution**. China (Taiwan).

**Diagnosis.** The species is similar to the new species *N. hujiayaoi*, and it may be distinguished from the latter by differently shaped patch of silvery pubescence on the elytral disc (Figure 5E) and shorter aedeagal paramere compared with median lobe (Figure 7E–H).


***Naddia hujiayaoi* Xia, Tang & *Schillhammer* sp. nov.**


**Type material. Holotype. China: Zhejiang:** ♂, glued on a card with labels as follows: “China: Zhejiang A. R., Lin’an City, Mt. West Tianmu, alt. 1100 m, 30 May 2014, Liang Tang leg.” “Holotype/*Naddia hujiayaoi*/Xia & Tang & Schillhammer” [red handwritten label] (SHNU). **Paratypes. Zhejiang:** 1♂, Pan’an County, Dapanshan N. R., alt. 1100 m, 23 April 2011, Su-Jiong Zhang leg. (SHNU); 1♂, Lin’an County, Mt. Tianmushan, alt. 300 m, 17 May 2006, Hu & Tang leg. (SHNU); 1♀, Lin’an City, Mt. Tianmu, 15–28 August 2003, Hu & Tang leg. (SHNU); 1♀, Lin’an County, Mt. Tianmushan, alt. 300 m, 17 May 2006, Hu & Tang leg. (SHNU); 1♀, Anji City, Longwangshan Mt., Qianmutian, 4.8 km, alt. 1050–1250 m, 08 June 2012, Hu & Yin leg. (SHNU); 1♀, Wuyanling, alt. 700 m, 09 May 2004, Hu, Tang & Zhu leg. (SHNU); **Fujian:** 1♂, Meihua Mt., Guihe Vill, alt. 1500 m, 20 May 2007, Huang & Xu leg. (SHNU); 1♂, Wuyishan City, Guadun Vill, alt. 1100–1400 m, 29 May 2012, Peng & Dai leg. (SHNU); 1♀, Meihua Mt., Guihe Vill, alt. 1500 m, 24 May 2007, Huang & Xu leg. (SHNU); **Jiangxi:** 1♀, Mt. Wuyi, alt. 950 m, 10 May 2005, Hu & Tang leg. (SHNU); 1♂1♀, Sanqingshan, 28°52′ N, 118°03′ E, alt. 600 m, 18 April–17 July 2007, J. Turna leg. (NMW); **Hunan:** 1♂, Yanlin County, Taoyuandong N. R., Nanfengmian, alt. 1350 m, 14 June 2016, Jiang, Liu, Jiang & Zhou leg. (SHNU); 1♂, 1♀, Shunhuang Shan For. Park, 26°23′ N, 111°01′ E, alt. 1480 m, 23 May–20 June 2013, Jatua leg. (NMW); **Guangxi:** 2♂♂, 1♀, Mao’er Shan, 25°55′ N, 110°28′ E, alt. 1500–1600 m, 5–19 June 2011, J. Turna leg. (NMW); **Guangdong:** 1♂, Yunji Shan Nat. Res., alt. 700–1300 m, 24°06–07′ N, 114°10′ E, 13–23 June 2013, Jatua leg. (NMW). All paratypes with yellow handwritten label: Paratype/*Naddia hujiayaoi*/Xia, Tang & Schillhammer”.

**Description.** Measurements of male: BL: 13.3–16.1 mm, FL: 7.5–8.9 mm. HL: 2.50–2.84 mm, HW: 2.67–3.06 mm, CL: 0.72–0.89 mm, PO: 1.33–1.61 mm, PL: 2.50–2.95 mm, PW: 2.39–2.78 mm, EL: 2.78–3.34 mm, EW: 3.00–3.61 mm. HW/HL: 1.07–1.11, PO/CL: 1.71–2.23, PL/PW: 1.00–1.08, EL/EW: 0.92–0.97.

Measurements of female: BL: 13.3–15.0 mm, FL: 7.2–8.3 mm. HL: 2.22–2.78 mm, HW: 2.56–3.11 mm, CL: 0.67–0.78 mm, PO: 1.33–1.56 mm, PL: 2.50–2.78 mm, PW: 2.45–2.84 mm, EL: 2.78–3.06 mm, EW: 2.95–3.28 mm. HW/HL: 0.94–1.15 mm, PO/CL: 1.93–2.00, PL/PW: 0.98–1.04, EL/EW: 0.93–0.98.

Body blackish, elytra blackish with golden, silver and black pubescence, lateral portions of abdominal tergites III–V with silver pubescence, tergite VII with a band of silver pubescence in basal half, mouthparts and tarsi brownish, antennae blackish, apical third of terminal segment brownish.

Head with dorsal surface densely punctate, punctation more or less confluent; ventral surface densely punctate with large portions of posterior angles impunctate; antennae with club flattened, antennomeres 1–3 distinctly oblong, antennomere 2 shorter than antennomere 3, antennomere 4 almost as long as wide, antennomeres 5–10 becoming gradually wider than long, antennomere 11 longer than penultimate, antennomeres 7–10 slightly asymmetrical, antennomere 11 asymmetrical.

Pronotum widest at anterior angles, anterior angles distinctly prominent; dorsal surface with punctures similar to those of head; an impunctate midline present in posterior fourth.

Elytra with lateral margins slightly divergent posteriad, punctation very dense and fine (Figure 8A,B). Scutellum densely and finely punctate, pubescence black.

Abdominal tergites III–VI each with punctation of median portion gradually becoming sparser apicad.

Male. Sternite VII slightly emarginate in middle of posterior margin; sternite VIII rounded emarginate in middle of posterior margin; 9th sternite triangularly emarginate in middle of posterior margin; aedeagus (Figure 9A–D) symmetrical, median lobe elongated, inner sac with weakly sclerotized sclerites; paramere slender, distinctly shorter than median lobe, slightly curved ventrad in lateral view, with few apical setae.

Female. Sternite VIII with posterior margin entire.

**Distribution**. China (Zhejiang, Fujian, Jiangxi, Hunan, Guangxi, Guangdong).

**Diagnosis.** The new species is similar to *N. monticola* Shibata, 1994 in general appearance, but it may be distinguished from the latter by continued elytral pubescence patch at middle of posterior margin (interrupted in *N. monticola*) and longer aedeagal paramere compared with median lobe (Figure 9A–D).

**Etymology.** This species is named in honor of our dear colleague Dr. Jia-yao Hu, who is working on Staphylinidae and collected some specimens of the new species.


**
*Naddia hainanensis*
**
**Yang & Zhou, 2010**


*Naddia hainanensis* Yang & Zhou, **2010**, 7.

**Material examined**. None.

**Distribution**. China (Hainan).


**Key to Chinese species of *Naddia***
Elytra with distinct bronze or coppery tint, elytral surface covered with reddish pubescence; segments of antennal club more cylindrical, last antennal segment elongate and symmetrical; species with large body size: BL = 16.4–23.6 mm..........................................................................................................................................2–Elytra without bronze or coppery tint, elytral surface without reddish pubescence (exception: *N. rufipennis*); segments of antennal club more flattened, last antennal segment mostly short and asymmetrical (exception: *N. hainanensis*); species with varied body size: BL = 13.3–20.0 mm........................................6Full-winged species, posterior margin of abdominal tergite VII with distinct palisade fringe................................................................................................3–Brachypterous species, posterior margin of abdominal tergite VII without or with indistinct palisade fringe................................................................5Aedeagal paramere with short setae..............................................................4–Aedeagal paramere with long setae (Figure 2A–D)). China (Yunnan), Myanmar, Laos.....................................................................................***N. miniata***Antennal segments 8–11 about as long as wide (Figure 8D). China (Beijing, Gansu, Zhejiang, Jiangsu, Guizhou, Sichuan, Henan, Hunan, Shaanxi, Fujian, Jiangxi, Chongqing, Guangxi, Hubei).........................................................***N. chinensis***–Antennal segments 8–11 shorter than wide (Figure 8E). China (Taiwan)..............................................................................***N. taiwanensis***First four visible tergites each with longitudinal golden pubescent patches laterally; aedeagal paramere sharply prominent with long apical setae. China (Guangxi)........................................................................***N. chenchangchini* sp. nov.**–First three visible tergites each with longitudinal golden pubescent patches laterally; aedeagal paramere roundly prominent with short subapical setae. China (Fujian, Guangdong, Hunan)..................................***N. mangshanensis***Elytra reddish; pronotum with transverse patch of dense golden pubescent midlength. China (Yunnan), Myanmar.................................................***N. rufipennis***–Elytra blackish; pronotum without patch of distinct golden pubescence............7Smaller species with BL = 11.7 mm, tibiae yellow. China (Hainan)......................................................................................***N. hainanensis***–Larger species with BL = 13.3–20.0 mm, tibiae black to dark brown............. 8Full-winged species, posterior margin of abdominal tergite VII with distinct palisade fringe; apical margin of terminal antennal segment not or slightly concave, forming a round tip....................................................................................9–Brachypterous species, posterior margin of abdominal tergite VII with indistinct palisade fringe; apical margin of terminal antennal segment strongly concave, forming an acute tip (Figure 8H). China (Guangdong, Hunan)......................................................................................***N. nanlingensis***Abdominal tergite VII with a median basal patch of silver pubescence; male sternite VII with large tufts of setae at basal median portion.........................................10–Abdominal tergites VI and VII each with a median basal patch of silver pubescence; male sternite VII without large tufts of setae...................................11Pubescence patch of each elytron continued at middle of posterior margin (Figure 8A,B); aedeagus with long paramere (Figure 9A–D). China (Zhejiang, Fujian, Jiangxi, Hunan, Guangxi, Guangdong).................................***N. hujiayaoi* sp. nov.**–Pubescence patch of each elytron interrupted at middle of posterior margin (Figure 5E,F); aedeagus with short paramere (Figure 7E–H). China (Taiwan)..................................................................................***N. monticola***Segments of antennal club more symmetrical (Figure 1 in Shibata 1994). China (Taiwan).............................................................................................................***N. ishiharai***–Segments of antennal club more asymmetrical (Figure 8I). China (Zhejiang, Jiangxi, Shaanxi, Guangxi, Jiangsu, Hunan, Hubei, Sichuan, Guizhou, Fujian, Guangdong), Vietnam................................................................***N. atripes***


## 4. Discussion

The genus *Naddia* is a southeast palearctic/oriental element. With the decrease of latitude, the species has become more numerous, at least on mainland Asia. It also occurs on the Sunda Islands, where it seems particularly rich on the island of Borneo. The scarcity of species on Sumatra and Java is most likely a result of bad sampling coverage and lack of proper collecting methods. In the Chinese fauna, most *Naddia* species occur in the southern half. Presently, Hunan is the most species-rich province, with five recorded species. However, considering the unidentified species in various collections, Yunnan, Guangxi and Guangdong have higher species diversity than Hunan. *Naddia chinensis* is the most widely distributed species in China. By reaching Beijing, Gansu and Shaanxi, this makes it the northernmost species of the genus.

The biology of the genus is almost unknown. According to our observations in the field, *Naddia* species are mostly diurnal insects. They are found in the light only very exceptionally. The most efficient methods for collecting them are flight intercept traps for the winged species and pitfall traps for wingless species.

Many species of *Naddia* are obviously excellent wasp imitators. They can be spotted crossing the path in forests during the day time. Their crawling behaviors mimic those of sphecid or vespid wasps: antennae pointed forward, repeatedly running for a short distance and stopping, shaking abdomen with high frequency (see Appendix A). *Naddia chinensis*, the only species observed in the field flying, mimics the flight behavior of sphecid or vespid wasps with abdomen hanging below when it is flying. The same applies to their morphological appearance. The first three abdominal segments of *Naddia* are often blackish with patches of golden pubescence laterally (Figure 10A–D). These patches of golden pubescence gradually become smaller apicad, and as a contrast, the median blackish parts gradually become larger apicad. Such combination may provide an illusion to predators that *Naddia* species have a slender waist as that of wasps. This imitation is especially effective in the forests, where the light is less strong.

## 5. Conclusions

The species of *Naddia* have excellent visual sense. Detecting the approach of the collector, they usually tend to catalepsy for a while (Figure 10D). The time span of catalepsy can be pretty long at lower temperature and vice versa. If the catalepsy strategy fails, they run fast to find a shelter. In this case, species with smaller size sometimes may also try to fly away, but this has never happened in *N. chinensis*.

## Figures and Tables

**Figure 1 insects-13-00503-f001:**
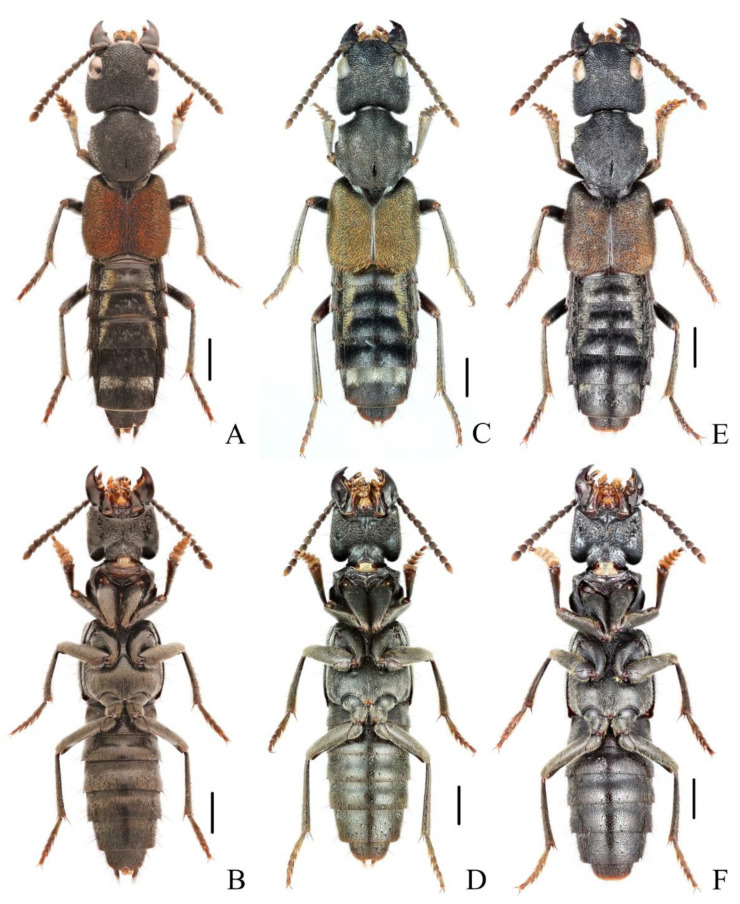
(**A**–**F**) Adult habitus of *Naddia*. (**A**,**B**) *N. miniata*; (**C**,**D**) *N. chinensis*; (**E**,**F**) *N. taiwanensis*. Scale bars = 2 mm.

**Figure 2 insects-13-00503-f002:**
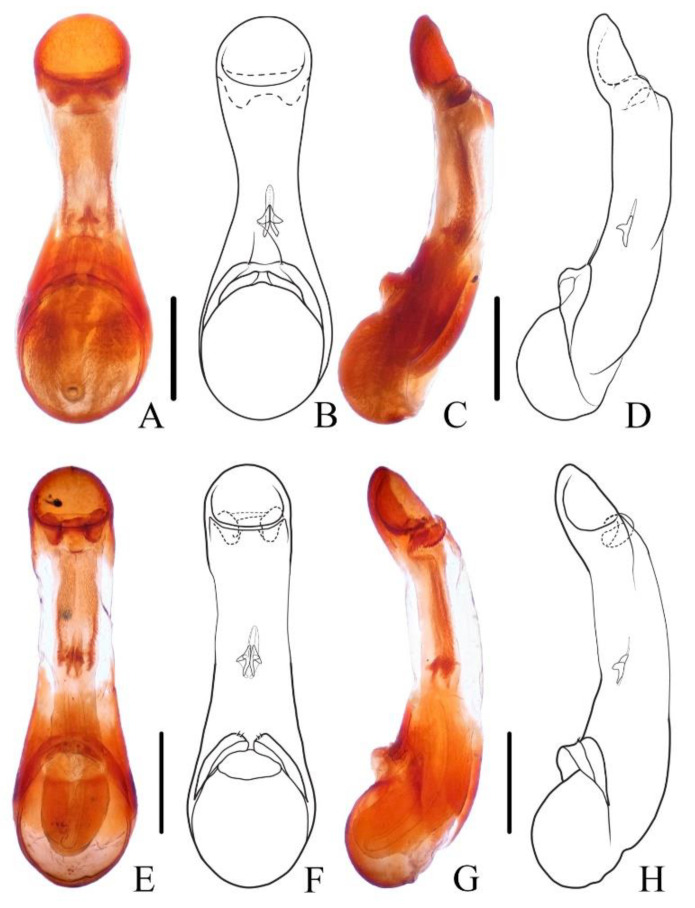
(**A**–**H**) Aedeagus of *Naddia*. (**A**–**D**) *N. miniata*. (**A**,**B**) ventral view; (**C**,**D**) lateral view. (**E**–**H**) *N. chinensis*. (**E**,**F**) ventral view; (**G**,**H**) lateral view. Scale bars = 0.5 mm.

**Figure 3 insects-13-00503-f003:**
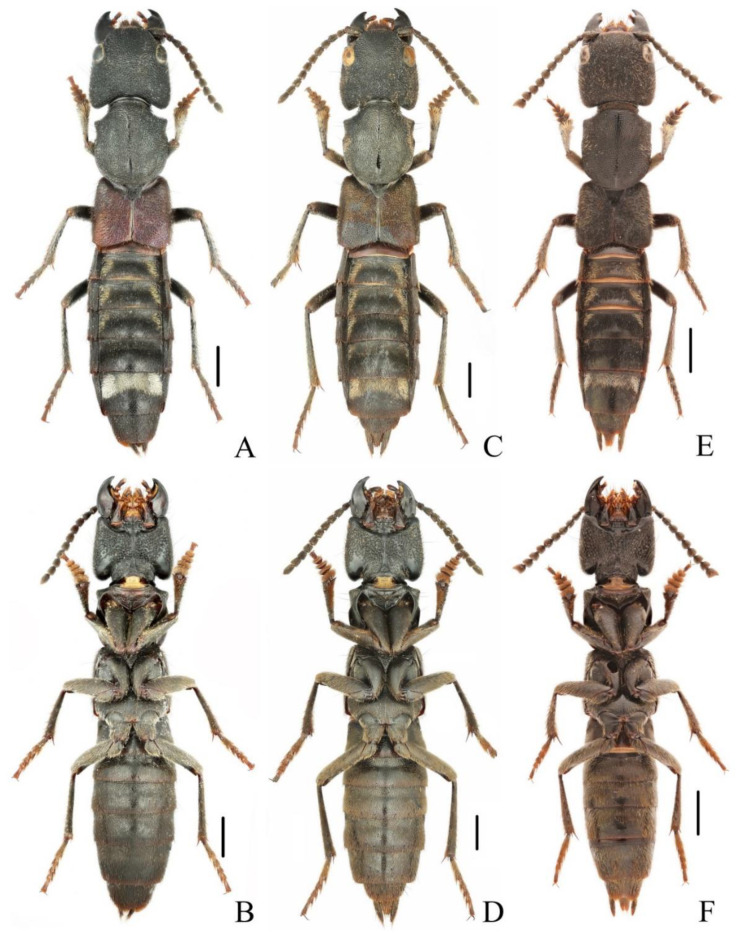
(**A**–**F**) Adult habitus of *Naddia*. (**A**,**B**) *N. mangshanensis*; (**C**,**D**) *N. chenchangchini* (paratype); (**E**,**F**) *N. nanlingensis*. Scale bars = 2 mm.

**Figure 4 insects-13-00503-f004:**
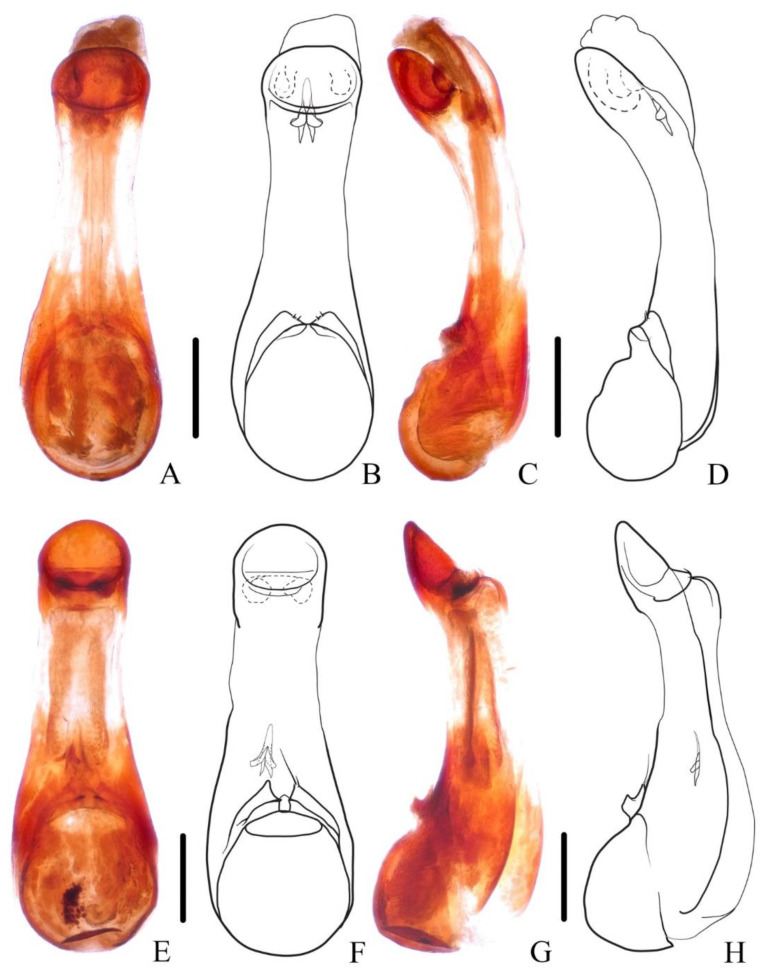
(**A**–**H**) Aedeagus of *Naddia*. (**A**–**D**) *N. mangshanensis*. (**A**,**B**) ventral view; (**C**,**D**) lateral view. (**E**–**H**) *N. chenchangchini*. (**E**,**F**) ventral view; (**G**,**H**) lateral view. Scale bars = 0.5 mm.

**Figure 5 insects-13-00503-f005:**
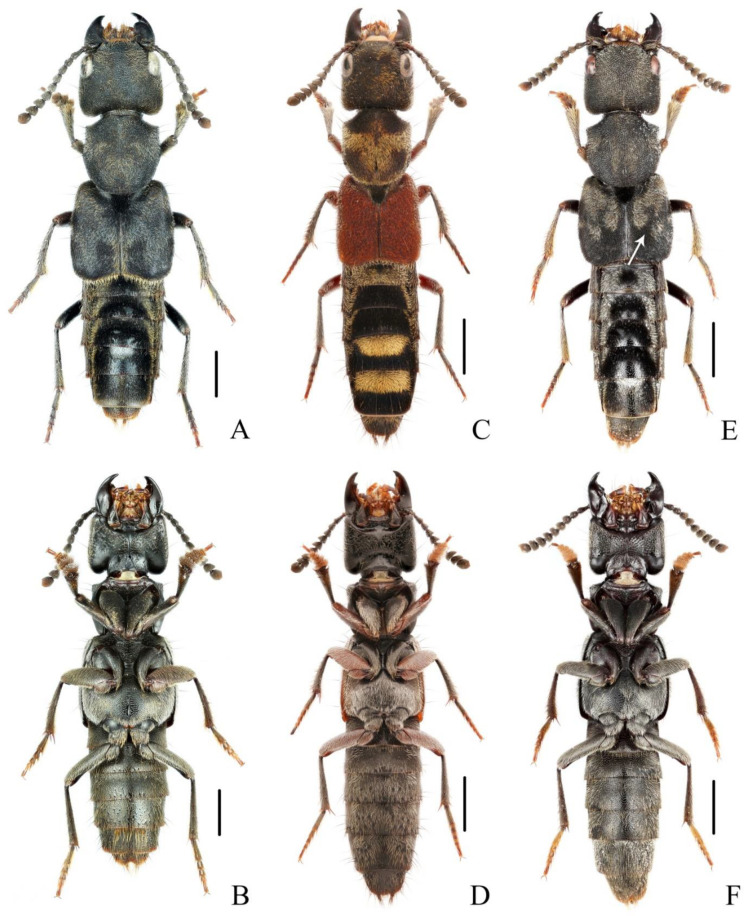
(**A**–**F**) Adult habitus of *Naddia*. (**A**,**B**) *N. atripes*; (**C**,**D**) *N. rufipennis*; (**E**,**F**) *N. monticola*. Scale bars = 2 mm.

**Figure 6 insects-13-00503-f006:**
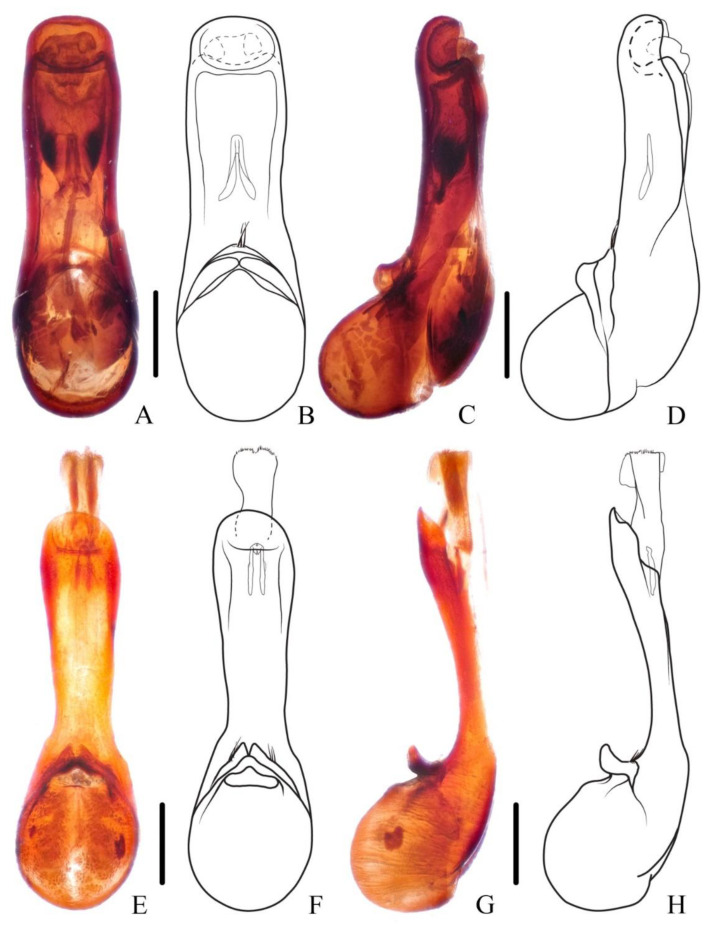
(**A**–**H**) Aedeagus of *Naddia*. (**A**–**D**) *N. nanlingensis*. (**A**,**B**) ventral view; (**C**,**D**) lateral view. (**E**–**H**) *N. atripes*. (**E**,**F**) ventral view; (**G**,**H**) lateral view. Scale bars = 0.5 mm.

**Figure 7 insects-13-00503-f007:**
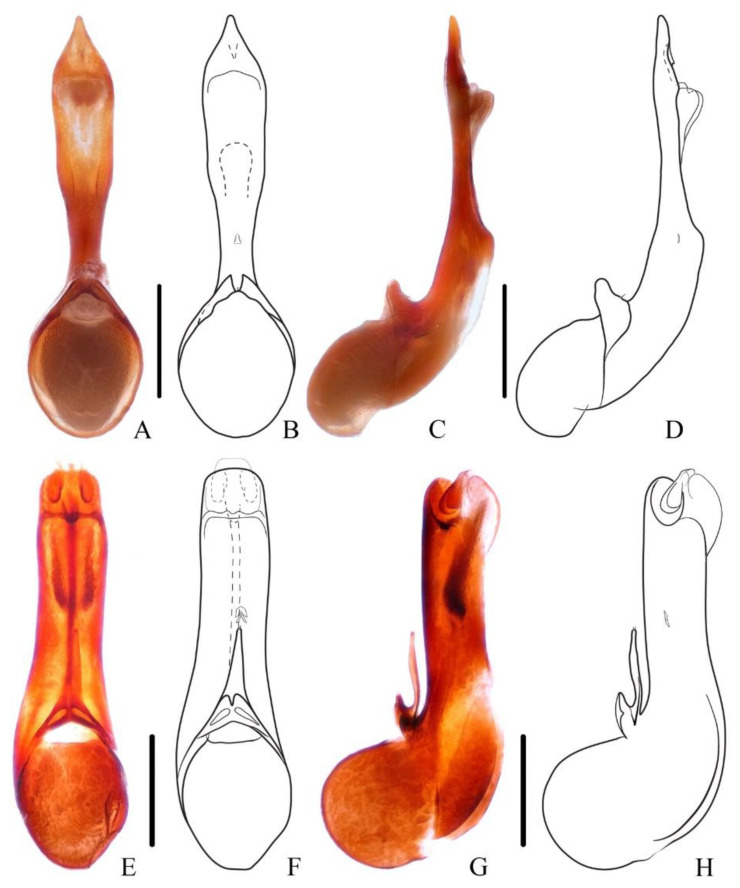
(**A**–**H**) Aedeagus of *Naddia*. (**A**–**D**) *N. rufipennis*. (**A**,**B**) ventral view; (**C**,**D**) lateral view. (**E**–**H**) *N. monticola*. (**E**,**F**) ventral view; (**G**,**H**) lateral view. Scale bars = 0.5 mm.

**Figure 8 insects-13-00503-f008:**
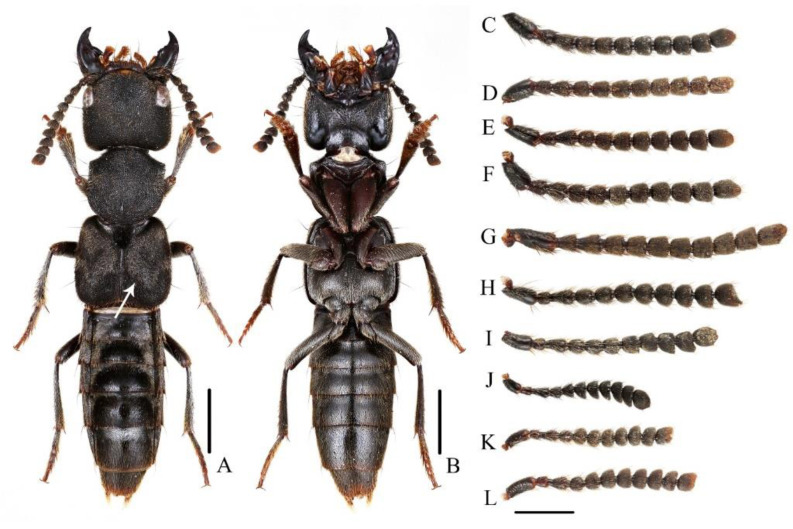
(**A**,**B**) Adult habitus of *N. hujiayaoi* (holotype). (**C**–**L**) antennae of *Naddia*. (**C**) *N. miniata*; (**D**) *N. chinensis*; (**E**) *N. taiwanensis*; (**F**) *N. mangshanensis*; (**G**) *N. chenchangchini*; (**H**) *N. nanlingensis*; (**I**) *N. atripes*; (**J**) *N. rufipennis*; (**K**) *N. monticola*; (**L**) *N. hujiayaoi*. Scale bars: 2 mm (**A**,**B**), 1 mm (**C**–**L**).

**Figure 9 insects-13-00503-f009:**
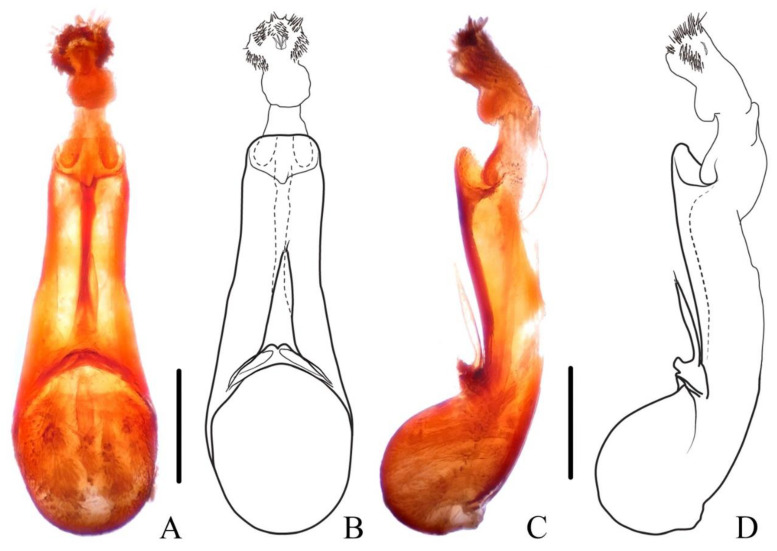
(**A**–**D**) Aedeagus of *Naddia hujiayaoi*. (**A**,**B**) ventral view; (**C**,**D**) lateral view. Scale bars = 0.5 mm.

**Figure 10 insects-13-00503-f010:**
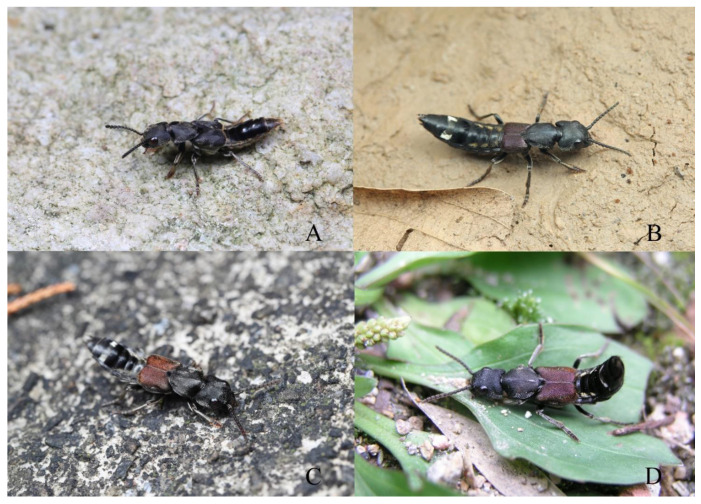
(**A**–**D**) Living *Naddia* species. (**A**) *N. atripes* (Photo by Liang Tang from Zhejiang, Longquan City, Mt. Fengyang at 10.v.2019); (**B**) *N. mangshanensis* (Photo by Liang Tang from Guangdong, Ruyuan County, Nanling N. R. at 20.v.2021); (**C**) *N. chinensis* (Photo by Liang Tang from Zhejiang, Lin’an City, Mt. West Tianmu at 18.viii.2019); (**D**) *N. chinensis* (Photo by Liang Tang from Zhejiang, Anji City, Mt. Longwang at 28.iii.2005).

## Data Availability

Data is contained within the article and Appendix A. The findings of this study are available from the corresponding author.

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
