# Peer review of "A Taxonomic Study on the Genus Naddia from China (Coleoptera, Staphylinidae, Staphylininae) with Descriptions of Two New Species †"

_insects, 2022, doi:10.3390/insects13060503_

Round 1

Reviewer 1 Report

My remarks and corrections of some places in the manuscript:

Page 3, line72. “Collction…” should be changed to “Collection”

Page 3, lines 83, 84. To remove “;”

Page 5, line 113. “N. chenchangchini”. Is it photograph of holotype or paratype? I recommend to note it here.

Page 7, line 120. “N. hujiyaoi”. See comment above.

Page 11, lines 241-244. To add type label for the paratypes. Example: All paratypes with red handwritten label:  Paratype / Naddia chenchangchini / Xia, Tang & Schillhammer”.

Page 19, lines 423-438. See comment above.

Page 22, line 516. “…pubescenceat” should be changed too “…pubescent”?

Page 24, line 602. “Bulletion…” should be changed to “Bulletin”…

Page 24, line 622. “Catalog…” should be changed to “Catalogue...”.

Author Response

All your modification suggestions have been received which are very useful to us and we strictly improved the manuscript by following the comments of you. Thanks a lot for the helps.

Page 3, line72. “Collction…” should be changed to “Collection”

We accept. It has been changed to “Collection”.

Page 3, lines 83, 84. To remove “;”

We accept. They have been removed.

Page 5, line 113. “N. chenchangchini”. Is it photograph of holotype or paratype? I recommend to note it here.

The photograph of “N. chenchangchini is paratype. It has been added there.

Page 7, line 120. “N. hujiyaoi”. See comment above.

The photograph of “N. hujiayaoi” is holotype. It has been added there.

Page 11, lines 241-244. To add type label for the paratypes. Example: „All paratypes with red handwritten label:  Paratype / Naddia chenchangchini / Xia, Tang & Schillhammer”.

We accept. It has been revised as follow in the paper: All paratypes with yellow handwritten label: Paratype / Naddia chenchangchini / Xia, Tang & Schillhammer”.

Page 19, lines 423-438. See comment above.

We accept. It has been revised as follow in the paper: All paratypes with yellow handwritten label: Paratype / Naddia hujiayaoi / Xia, Tang & Schillhammer”.

Page 22, line 516. “…pubescenceat” should be changed too “…pubescent”?

We accept. It has been changed to “…pubescent”.

Page 24, line 602. “Bulletion…” should be changed to “Bulletin”…

We accept. It has been changed to “Bulletin”.

Page 24, line 622. “Catalog…” should be changed to “Catalogue...”.

We accept. It has been changed to “Catalogue”.

Reviewer 2 Report

The work faces the revision of a genus of Staphylinidae which includes many species.
The originality of the argument concerns precisely the fact of revising a genus for the first time and describing new species for science and providing a key for identifying taxa.
The work is therefore extremely useful in the field of taxonomy and systematics.
The text is excellent, clear and easy to read and concerning English language, as far as I am concerned and within my abilities, nothing to complain.
The conclusions are consistent with the purpose of the work. I would add that the photos and drawings are also of excellent quality and the bibliography is complete.

Author Response

Thanks a lot for the reviewing and for your recognition.